# Effects of Seed-Layer N_2_O Plasma Treatment on ZnO Nanorod Based Ultraviolet Photodetectors: Experimental Investigation with Two Different Device Structures

**DOI:** 10.3390/nano11082011

**Published:** 2021-08-05

**Authors:** Seungmin Lee, Kiyun Nam, Jae Hyun Kim, Gi Young Hong, Sam-Dong Kim

**Affiliations:** Division of Electronics and Electrical Engineering, Dongguk University, Seoul 100715, Korea; seungminlee@dgu.ac.kr (S.L.); kynamkr@gmail.com (K.N.); kjh970612@naver.com (J.H.K.); e150great@naver.com (G.Y.H.)

**Keywords:** ZnO nanorods, ultraviolet photodetectors, N_2_O plasma treatment, high electron mobility transistor, interdigitated electrode

## Abstract

The crystalline quality of ZnO NR (nanorod) as a sensing material for visible blind ultraviolet PDs (photodetectors) critically depends on the SL (seed layer) material of properties, which is a key to high-quality nanocrystallite growth, more so than the synthesis method. In this study, we fabricated two different device structures of a gateless AlGaN/GaN HEMT (high electron mobility transistor) and a photoconductive PD structure with an IDE (interdigitated electrode) pattern implemented on a PET (polyethylene terephthalate) flexible substrate, and investigated the impact on device performance through the SL N_2_O plasma treatment. In case of HEMT-based PD, the highest current on-off ratio (~7) and spectral responsivity ***R*** (~1.5 × 10^5^ A/W) were obtained from the treatment for 6 min, whereas the IDE pattern-based PD showed the best performance (on-off ratio = ~44, ***R*** = ~69 A/W) from the treatment for 3 min and above, during which a significant etch damage on PET substrates was produced. This improvement in device performance was due to the enhancement in NR crystalline quality as revealed by our X-ray diffraction, photoluminescence, and microanalysis.

## 1. Introduction

UV (ultraviolet) light can be classified in a variety of ways, but it can be generally divided into vacuum UV (VUV) 200–10 nm, deep UV (DUV) 350–190 nm, UV-A 400–320 nm, and UV-B 320–280 nm with respect to the range of wavelength. With the rapid development of aerospace technology, intensive effort for UV measurement in the stratosphere has been initiated, and since then, interest for high-sensitivity sensing has been amplified in medicine, environment, health, communication, military, space science, and a wide range of fields [1,2]. The main technical requirements for high-performance UV PDs (photodetectors) are as follows: (i) not responding to light in the visible range (visible blind), (ii) high quantum efficiency and responsivity, (iii) fast response speed, (iv) low-level base signal and noise, and (v) wide detection range in wavelength [3]. The development of UV PDs has been driven by two main streams: a photoemissive sensor and a semiconductor sensor, and the semiconductor-based sensors have many technical advantages, such as broadband responsivity, excellent linearity, high quantum efficiency, and compact device size, even though they normally exhibit a slower response.

ZnO nanostructures in general exhibit a much higher photoresponse than ZnO film under UV radiation owing to the increased surface-to-volume ratio, nanowire carrier transport channel, and much higher exciton binding energy [1], which enables ZnO NR (nanorod) arrays to be more efficient in generating electron-hole pairs under UV illumination. Many research efforts have been made to improve the performance of the ZnO NR-based UV PDs fabricated on semiconductors/glasses or flexible substrates [4,5] in a variety of ways, such as crystalline quality improvement of NRs [6,7,8]; utilization of the hybrid structure of ZnO nanocrystals with other materials, such as copper oxide [9] and graphene [10], and decoration of Ag nanoparticles on the ZnO NR surface [11]. However, most of the passive type PDs reported to date exhibit a very slow response speed (tens of seconds) and low responsivity (hundreds of A/W) due to their poor material quality and bottleneck sensing reaction based on the adsorption/desorption kinetics of O_2_ (oxygen) on ZnO surface.

We investigated the effect of N_2_O plasma treatment for the SL (seed layer) using two different device structures: a photoconductive PD with IDE (interdigitated electrode) patterns implemented on PET (polyethylene terephthalate) flexible substrate, and an active PD fabricated by growing ZnO NRs on the gate region of GaN/AlGaN HEMT (high-electron-mobility-transistor) structure. These PDs had completely different device structures, sensing principles, and processing methods and temperature, except for the fact that the UV absorbing layer was made of the same material. N_2_O molecules are very stable at room temperature; however, N–O bonds of N_2_O in cold plasma, which are relatively weaker than N–N bonds, can be broken easily by producing various plasma radicals. These radicals can critically influence the electrical and optical properties of as-grown ZnO crystallites; for example, in the case of bottom-gated ZnO thin film transistors, the on-off current ratio was improved by two orders of magnitude by postfabrication annealing and subsequent N_2_O plasma treatment due to the reduction in oxygen vacancies at the top region of the ZnO channel [12]. We examined, in this study, growth morphologies, optical properties, and structural evolution of the NRs depending on treatment condition. Performance parameters of each fabricated PD device were also assessed in order to substantiate our material characterizations for the ZnO crystals.

## 2. Experimental Section

Shown in Figure 1a,d are the schematics of two different device platforms employing the ZnO NRs as a UV-sensing material fabricated in this study. We also showed the schematic illustrations of a major process flow for each device in Figure 2. HEMT-based PDs (Figure 1a) comprised the epitaxial structure of the GaN buffer/3000-nm GaN channel/20-nm Al_0.25_Ga_0.75_N barrier/1.25 nm GaN cap from the bottom which were grown on (111) Si substrate by using metal organic chemical vapor deposition. After a mesa formation (100 × 40 µm^2^), the ohmic contacts (30/180/40/150 nm Ti/Al/Ni/Au) were achieved by the pattern liftoff and subsequent rapid thermal annealing at 950 °C for 30 s. To complete the gateless HEMT structure, a floating gate region (2 × 100 µm^2^) was then defined by selective CF_4_ plasma etching the 100 nm Si_3_N_4_ passivation layer covered onto the device. On the other hand, the IDE (interdigitated electrode) pattern on a flexible PET (polyethylene terephthalate) shown in Figure 1d was fabricated in the following way. After cleaning the PET substrates, O_2_ plasma treatment was performed on the substrate for 5 min to change its hydrophobic nature to hydrophilic to ensure a good adhesion of the metal layers to be grown in the next step (50 sccm O_2_ flow rate, 100 W RF power, 50 mTorr chamber pressure). An IDE pattern (width/space of 10/2 μm with 20 fingers) using Ti/Au metal stacks (30/80 nm) was formed with an active device area of 0.295 mm^2^ on commercially available PET substrates by using photolithography and pattern liftoff.

We used an aqueous solution-based hydrothermal method for the ZnO NR array synthesis on each device structure. First, SL solution (20 mM concentration colloidal solution of zinc acetate dehydrate [Zn(CH_3_COO)_2_·2H_2_O] in n-propyl alcohol [C_3_H_8_O]) was spun onto the patterned devices at 3000 rpm for 30 s, and they were then annealed at 100 °C for 60 s. This coating step was repeated 15 times to achieve a final thickness of ~20 nm as observed in cross-sectional transmission electron microscopy (TEM) of Figure 3. The coated SLs were postannealed at 300 and 150 °C for the gateless HEMT and IDE pattern-based device, respectively, for 60 min. Before the growth of NRs, the SL coated substrates were exposed to N_2_O plasma in a reactive ion etching system at a gas flow rate of 50 sccm, a pressure of 50 mTorr, and RF power of 100 W. To examine the effect of the plasma exposure, four different samples were prepared for each device type: one with no plasma treatment for reference (CS, control sample), and three others of different plasma exposure time to the SLs, for 1, 3, and 6 min. In the case of HEMT-based PDs, the SL-coated samples were patterned with photoresist to cover the gate region for the subsequent etching of the SLs from all the areas except the gate region. The NRs were grown in the same condition such that all the samples were immersed in the growth solution (an equimolar 25 mM aqueous solution of zinc nitrate hexahydrate [Zn(NO_3_)_2_·6H_2_O, 99%] and hexamethylenetetramine [HMTA, C_6_H_12_N_4_, 99.5%]) at 90 °C for 6 hr. After completing the NR growths, we finally rinsed and dried all the samples by blowing N_2_. Each type of fabricated PD was shown in Figure 1b,e. Typical morphology of NR crystallites grown on unpatterned substrates are shown in Figure 1c. Crosslinked high-density NR arrays grown between the electrodes were obtained as shown in Figure 1f from the IDE pattern-based devices.

To examine the surface topography of SLs as well as the growth morphology, crystalline, and optical properties of as-grown NR crystals, the samples were grown on unpatterned substrates of (002) GaN and PET substrates under the different SL plasma exposures. These samples were characterized by a scanning electron microscope (SEM, S-4800S, Hitachi, Japan, at 10 kV) and a transmission electron microscope (TEM, H-9500, Hitachi, Japan, at 500 kV). The effects of plasma treatment on the surface morphology of SL films were also investigated by atomic force microscopy (AFM, N8-NEOS, Bruker, Germany) in noncontact mode before growing the NRs on each plasma treated SL. Near band edge emission (NBE) and intra-sub-band emissions associated with shallow or deep level defects of ZnO NRs were characterized by room temperature photoluminescence (PL, RPM2000, Nanometrics, Korea) by using a 325 nm illumination of He-Cd laser. Absorption spectrum analysis of ZnO NRs grown on PET substrates was carried out by using a UV-vis spectrophotometer (T-60, PG Instruments, Lutterworth, UK) at room temperature. The X-ray diffraction (XRD) measurements were also carried out to investigate the crystalline quality of NRs.

A Keithley SMU (source meter unit) was used to measure *I-V* characteristics of the fabricated devices under both dark and UV illumination conditions. The spectral response of the PDs was measured by using a monochromatic light source with a wavelength range from 290–700 nm. A broad band Xenon lamp (200–1100 nm, 300 W) light was filtered and chopped into monochromatic light using a computer program controlled diffraction grating. Uniform monochromatic light was passed through a frequency chopper wheel at 30 Hz and illuminated on to the sample. The frequency of monochromatic light was fed to a lock-in amplifier, and the photocurrents from the devices were recorded at each wavelength (290–700 nm) and analyzed.

## 3. Results and Discussions

As-coated SLs by sol-gel method at room temperature are amorphous in general, but they were converted to a crystalline state during the subsequent annealing steps [13]. Crystallization of ZnO thin films is known to be affected by various variables such as sol solution, annealing environment, temperature, and coating methods [14,15,16]. As shown in our cross-sectional TEM micrographs of Figure 3, the SLs coated on the GaN substrates by our sol method exhibited a nanocrystalline structure of few nanometers in grain size as observed in a local area. The surface roughness estimated by AFM RMS (root mean square) was ~4.5 nm in the case of untreated SL; however, the RMS roughness was continuously reduced to 3.2, 2.9, and 2.3 nm with the increase in plasma treatment time to 1, 3, and 6 min. This shows a good agreement with planar SEM views (insets in Figure 3) where more conformal and higher surface coverage SLs are shown after the plasma treatment for 6 min. On the other hand, the surface roughness of the SLs coated on PET substrates was increased with the plasma treatment time as observed in AFM micrographs of Figure 3c; particularly, a significant roughness of ~240 nm was obtained in the case of 6 min treatment. Considering the fact that the maximum surface roughness of bare PET substrate is only few tens of nanometers, this huge surface roughness suggests that the N_2_O plasma treatment started to etch the PET substrate through the vulnerable regions of the seed crystals, such as grain boundaries or the areas of minimum SL growth.

Microstructural evolution of the NR crystals depending upon the SL plasma treatment and substrate material was shown in Figure 4. NR crystallites grown on the GaN substrates showed a gradual reduction in average length but an increase in mean diameter with plasma treatment time, where average length and diameter in each case were estimated from 80 NRs grown on four different locations of 2 × 2 cm^2^ samples. The XRD patterns for the NRs grown on GaN substrates with the SLs prepared under different conditions were shown in Figure 5a. Single intense peak (2θ = 34.47°) from (002) reflection was observed from the NRs grown with the SLs plasma-treated for 6 min, and no other peak related to any different orientation was found. This reveals that the ZnO NR crystals grow dominantly along the c-axis in a vertical direction to the substrate under this condition. On the other hand, we obtained weaker (002) reflections with other visible peaks from (100), (101), and (102) planes at 31.7, 36.4, and 47.5º, respectively, from the NRs grown under different conditions. This enhancement in preferred orientation of the NR crystals can be associated with the promoted crystalline quality of SL crystals in terms of O/Zn stoichiometry, surface smoothness, and the re duction in OH and oxygenated carbons originated by seed solution [16]. We have an opposite trend of “increase” in crystal length with the treatment time up to 3 min in the case of NRs on PET substrates, which can be associated with the significant increase in surface roughness caused by plasma treatment. As the dominating nucleation mechanism for the NR arrays switches from “surface nucleation” to “grain-boundary nucleation” [17] with the increase in three-dimensional nucleation sites caused by rougher SL surface, (002) reflections were increased to the case of 1 min treatment but decreased with further longer treatment as shown in the XRD spectra. The aspect ratio (length/diameter) of the NR crystal continuously decreased from 18 to 4 with plasma treatment time when the growths were performed on GaN substrates; but, in the case of PET substrates, it increased up to 3 min treatment and then fell off as shown in Figure 4c,d. For the NR growths on PET substrates, three-dimensional growth by random nucleation, such as seedless growth, appeared to be intensified via severe etch damage against PET substrates by the plasma treatment over 3 min, as shown in the SEM micrograph of Figure 4b and XRD results. As a result, the length of the as-grown NRs was also suppressed after 3 min treatment, as shown in Figure 4d.

It is well understood that ZnO crystallites grown by hydrothermal methods include many hydroxyls and carbon functional group complexes near the surface, such as hydrocarbons (COOH groups) and carbon oxides (COOR) [18,19]. Because the N_2_O plasma radicals, such as NO*, N*, and O* [20], of high reactive energy can provide enough enthalpy for the reaction reducing the OH, COOH and COOR groups, the following reaction can take place during the N_2_O plasma treatment.
(1)$${\mathrm{COOH}}^{-}+{\mathrm{NO}}^{*}\;\to {\;N}_{2}O+{\mathrm{CO}}_{2}+H_{2}O\;$$
(2)$${\mathrm{COOH}}^{-}+O^{*}\;\overset{\;}{\to }\;\mathrm{CO}+{\mathrm{CO}}_{2}+H_{2}O\;$$
(3)$$\mathrm{Zn}{\left(\mathrm{OH}\right)}_{2}\;\overset{{\;N}_{2}O^{*}}{\to }\;\mathrm{ZnO}+H_{2}O\;$$

As shown in the PL spectra of Figure 6a,b, the NRs grown on plasma-treated SLs showed significantly enhanced near band emission (NBE) in a UV zone of 350–400 nm arising from excitonic band-to-band recombination in ZnO [6,20]. In the case of NRs grown on PET substrates, the NBE was also increased with treatment time. However, it reduced after 6 min treatment caused by the severe etching damage to the substrates. This could be due to the enhanced crystalline quality of SLs in terms of various characteristic changes, such as promoted structural integrity as revealed by XRD analysis and the suppression of intrinsic defects and/or impurities with the recovery of O/Zn stoichiometry. The impurities such as carbon functional groups have strong electron scavenging properties, therefore they lead to serious degradation in NBE and increase trap-assisted emission in each respective wavelength range [18,21]. Broadband (400–750 nm) visible PL emissions shown in Figure 6a,b are generally associated with various complicated forms of intrinsic defect states such as oxygen vacancies (***V_O_***), zinc vacancy, oxygen interstitial (***Zn_i_***), zinc interstitial, oxygen antisite, zinc antisite, and native defect clusters such as ***V_O_-Zn_i_*** [22]. Our understanding of the origins of the visible emissions with respect to different defect states is still not complete, and the exact nature of distinctive emission patterns centered at 460 nm (blue emission), 550 nm (green emission)), 580 nm (yellow emission), and 640 nm (red emission) is the subject of the most debate depending on the synthesis method. All these visible emissions were clearly suppressed in the case of GaN substrate by the SL treatment; however, emissions from the NRs on PET substrates were increased slightly in a region of 550–640 nm with plasma treatment, while blue emission (~460 nm) showed a rather more effective reduction upon treatment over 3 min. Figure 6c demonstrates normalized optical absorption spectra measured from the NRs measured by a UV—visible spectrometer at room temperature. All the NRs showed a maximum absorption at ~350 nm; however, this maximum intensity was dependent on the SL plasma treatment condition. The NR crystals grown on the SLs plasma-treated for 3 min showed the highest absorption, and this evolutionary change of absorption showed a good agreement with the PL spectra.

The effects of SL plasma treatment on the device performance of two different PD structures were examined by measuring their *I-V* characteristics, spectral response, and transient photoresponse. Each PD structure had its unique strength and weakness in performance. The HEMT-based PDs especially exhibited very high responsivities greater than 10^3^~10^4^ A/W due to their gain characteristics [23], however, they delivered a relatively low current on-off ratio. Photoconductive PDs with ZnO NR cross-linking arrays between planar electrodes implemented on plastic substrates generally showed smaller responsivities (few tens of A/W) but higher current on-off ratios [4,20]. The light sensing mechanism of the ZnO-based photodetectors upon UV irradiation was basically associated with the desorption process of chemisorbed atmospheric O_2_ on the ZnO NR surface [24,25]. As-grown ZnO nanostructures grown by hydrothermal scheme are unintentional n-type semiconductors in nature due to native point defects such as zinc interstitials and oxygen deficiencies [13]. Channel conductance of the two-dimensional electron gas in HEMT-based PD was controlled by the charge development on the surface of ZnO NR array in the floating gate region [26]. When UV light was illuminated, reduced channel conductance caused by negative O_2_^−^ ions adsorbed to the ZnO under dark state could be recovered by the O_2_ desorption. This desorption was induced by the recombination of photogenerated excess holes with O_2_^−^ ions on the ZnO surface, but this recombination process was retarded by the presence of defect states acting as free carrier trap centers, thereby delivering the reduced drain current ***I_on_*** (photocurrent). Dark-state channel conductance could also be increased by the surface states suppressing the adsorption of O_2_^-^ ions with the reduction in negative surface potential of ZnO [27], and this gave rise to increased ***I_off_*** (dark current). As summarized in the inset table of Figure 7(a1), the on-off current ratio (***I_on_***/***I_off_*** measured at a drain voltage ***V_ds_*** of 5 V) of the HEMT-based PDs was increased from 2.2 (in the case of CS) to 7.5 when the SL was plasma-treated for 6 min. This supports that any slight modification of ZnO surface states by plasma treatment can influence the photoresponse performance of this device. UV photocurrent ***I_on_*** and dark current ***I_off_*** measured from each PD fabricated under different SL plasma-treatment conditions are shown in Figure 8.

One vital figure of merit for the PDs is the spectral responsivity ***R*** given by
(4)$$R=\frac{QE\cdot \lambda }{1.24}\mathrm{CF}$$
where QE is the quantum efficiency measured at a wavelength of λ, and CF is a correction factor given by the dimensions of illuminated beam and device active area. ***R*** is also defined by (***I_on_*** − ***I_off_***)/***P_i_***, and ***P_i_*** is the radiant light power incident on the active area. Responsivities of the HEMT-based PDs were measured using an intensity of 16 μW/cm^2^ and an effective area of 5.76 mm^2^ at a ***V_ds_*** of 1 V, whereas an effective active area of 0.238 mm^2^ at 2 V biasing and an intensity of 140 µW/cm^2^ were used for IDE pattern-based PDs. A highest spectral responsivity ***R*** of ~1.5 × 10^5^ A/W was obtained from the PDs at a wavelength of 360 nm, as revealed in Figure 7, when plasma-treated for 6 min due to improvement in ZnO material quality.

The UV sensing mechanism of the IDE pattern-based PDs is based on the modulated photoconduction through the crosslinked NR–NR bridges caused by O_2_ adsorption and desorption on the NR surface [28]. In the dark state, expanded ZnO surface depletion caused by O_2_^-^ adsorption brings about a minimum cross-section of conducting area and leads to a small dark current flowing inside the NR. Upon UV illumination, the surface depletion layer will shrink with O_2_ desorption and result in an increased photocurrent. More than this dimensional effect of each NR crystal, the surface depletion regions in every NR crosslinking act as energy barriers, preventing the electron transfer through the NR–NR bridge. Supposing defects near the surface of NRs are considerably reduced by plasma treatment, this will result in lowering the surface band bending with the reduction in electron energy barrier at the NR–NR junction, as well as an increase in the conducting area inside the NR. Figure 7(a2) shows the measured spectral responses of IDE-based UV-PDs measured at a bias voltage of 2 V. Responsivity was increased with the plasma treatment time, and the maximum ***R*** of ~69 A/W was achieved at a wavelength of 370 nm in the case of treatment for 3 min. However, from the 6 min SL treatment, a reduced ***R*** of ~62 A/W was obtained due to degraded NR crystalline quality as discussed earlier. On-off current ratios measured at 2 V were summarized in the inset table, and the best ratio of ~44 was obtained in the case of 3 min treatment.

Realtime transient responses measured from each PD were shown in Figure 7(b1,b2). “Rise time” and “fall time” under UV illumination on and off were estimated by the time intervals for the photocurrent to rise up to 90% of the maximum saturation under UV turn-on and for the current to fall off by 90% from the maximum under UV turn-off, respectively. The response time measured from the HEMT-based PDs was not significantly affected by the plasma treatment, and the fall time was even slightly increased with treatment time as shown in Figure 7(b1). This is not fully understood yet, but it can be attributed to the reduced total surface area of ZnO NR array caused by a significant reduction in aspect ratio with plasma treatment (see Figure 4c). As proposed in an oxygen desorption kinetics model reported by Khan et al. [25], a light-absorbing surface area which controls the rate of gate charge reduction can be a critical influencer determining the response speed in HEMT-based detectors. On the other hand, the response time was improved with the treatment time up to 3 min in the case of IDE-based PDs as shown in Figure 7(b2). Plasma treatment can promote the transport rate of photogenerated carriers moving either inward or outward direction to the surface of ZnO NRs due to the defect reduction (suppressed trapping/detrapping transient time), and thereby accelerating the rate of shrinking or expansion of depletion layer on the NR surface.

Visible blind solid-state UV PDs that have entered commercialization or are close to it can be wide bandgap material based (SiC or GaN) Schottky or diode type devices. In the case of GaN-family based UV detectors, it was not quite commercially applicable until the breakthrough of metal-organic chemical vapor deposition in the 1990s. Recent technology buildup for the III-nitride materials could open more reliably, countering various applications, especially at high temperatures or under high energy radiation. SiC is also one of the most important wide band gap materials in the development of UV PDs, and SiC based PDs can achieve large gains, excellent thermal resistance, high signal-to-noise ratios, and good solar-blind operation. These devices in the bulk-dominant type exhibit very fast response time of few nano- to picoseconds, but quite low responsivities of less than 0.2 A/W [1,29,30]. The ZnO nanostructure-based PDs reported in this study showed relatively high responsivities but slow response times due to the sensing mechanism of O_2_ adsorption/desorption. The ZnO NR-based devices are expected to highlight their need for specific applications detecting very weak UV signals regardless of the speed, such as pollution monitoring, water purification, noninvasive tissue diagnostics, and missile plume detection.

## 4. Conclusions

N_2_O plasma treatments were applied to the SLs for ZnO NRs hydrothermally grown for the UV absorbing layers of PDs. The effects of the plasma treatment were examined by fabricating two different visible blind detectors of gateless AlGaN/GaN HEMT structure and IDE pattern-based photoconductive cells on PET substrates. Significant improved photoresponse characteristics were achieved in terms of on-off current ratio (~7.5 for HEMT PDs, ~44 for IDE PDs) and responsivity (~1.5 × 10^5^ A/W for HEMT PDs, ~69 A/W for IDE PDs) when the SLs were plasma-treated. This is due to the effective suppression of intrinsic defects and enhanced crystalline quality of NRs regardless of device structure as revealed by our material characterizations using XRD, PL, and microanalysis.

## Figures and Tables

**Figure 1 nanomaterials-11-02011-f001:**
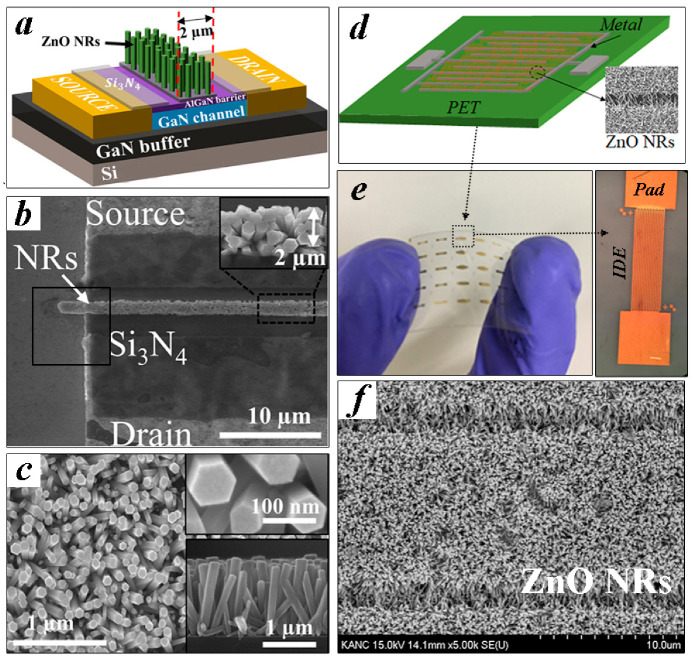
(**a**) Schematic illustration and (**b**) planar-view SEM micrograph of the fabricated ZnO NR-gated AlGaN/GaN HEMT. (**c**) Planar-view, (top-right inset) expanded top-view, and (bottom-right inset) cross-sectional SEM views of as-grown ZnO NRs. (**d**) Schematic illustration of IDE pattern-based UV PD fabricated on the PET substrate. (**e**) Fabricated IDE pattern-based UV PD (right inset: an expanded view of PD) with (**f**) a SEM micrograph showing the grown NR array on a metal pattern.

**Figure 2 nanomaterials-11-02011-f002:**
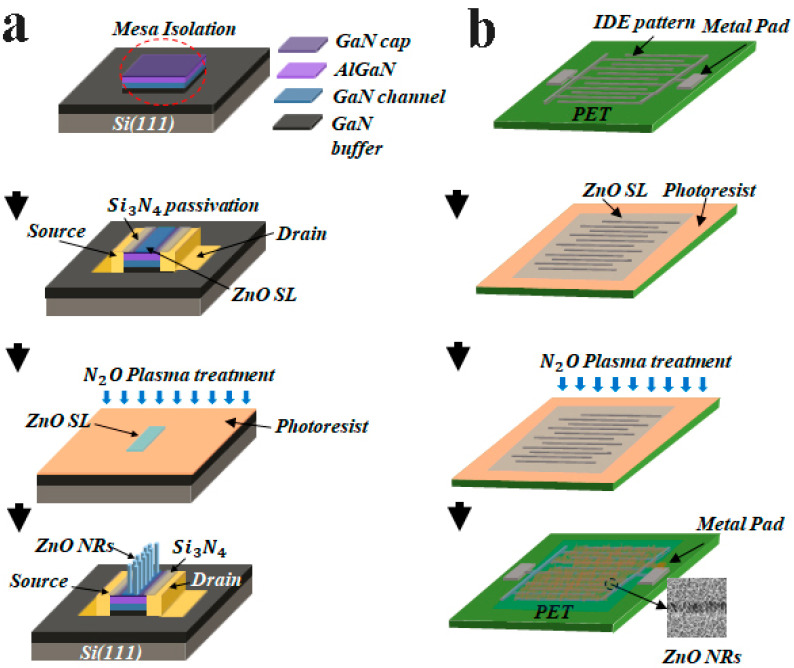
Schematic illustration of process flows for (**a**) HEMT-based and (**b**) IDE pattern-based PDs.

**Figure 3 nanomaterials-11-02011-f003:**
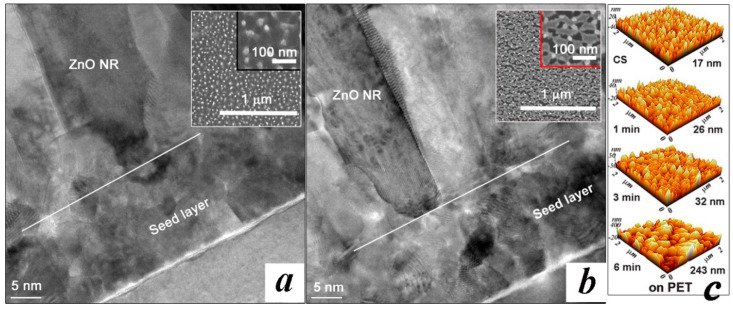
High-resolution cross-sectional TEM views of the ZnO NRs grown on the GaN substrates with (**a**) untreated SLs (CS) and (**b**) the SLs N_2_O plasma-treated for 6 min. Planar SEM views of the SLs prepared at each condition are shown in the top-right insets. (**c**) AFM surface morphologies of the untreated (CS) and N_2_O plasma treated SLs for 1, 3, and 6 min grown on PET substrates.

**Figure 4 nanomaterials-11-02011-f004:**
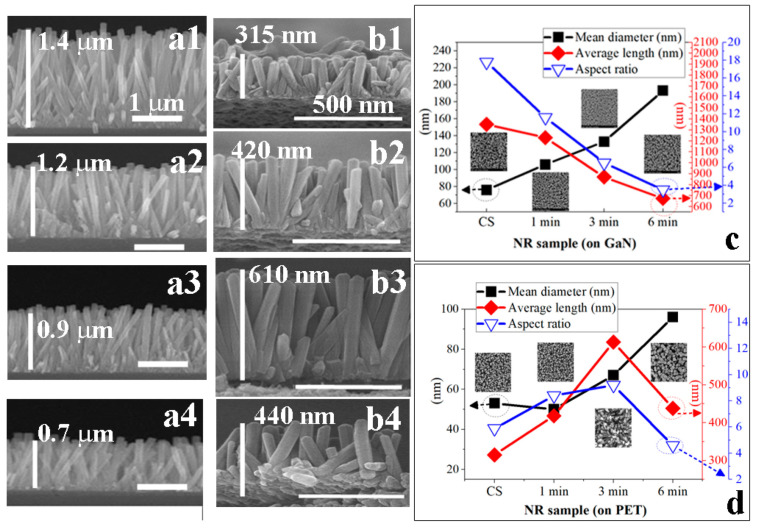
Cross-sectional SEM micrographs of the ZnO NRs grown with untreated SLs (CS, control sample) and the SLs N_2_O plasma-treated for 1, 3, and 6 min in the cases of growths on ((**a1**) CS, (**a2**) 1 min, (**a3**) 3 min, (**a4**) 6 min) the GaN and ((**b1**) CS, (**b2**) 1 min, (**b3**) 3 min, (**b4**) 6 min) PET substrates. Summary of mean diameters, average lengths, and aspect ratios (length/diameter) of ZnO NRs grown under different SL treatment conditions on (**c**) the GaN and (**d**) PET substrates.

**Figure 5 nanomaterials-11-02011-f005:**
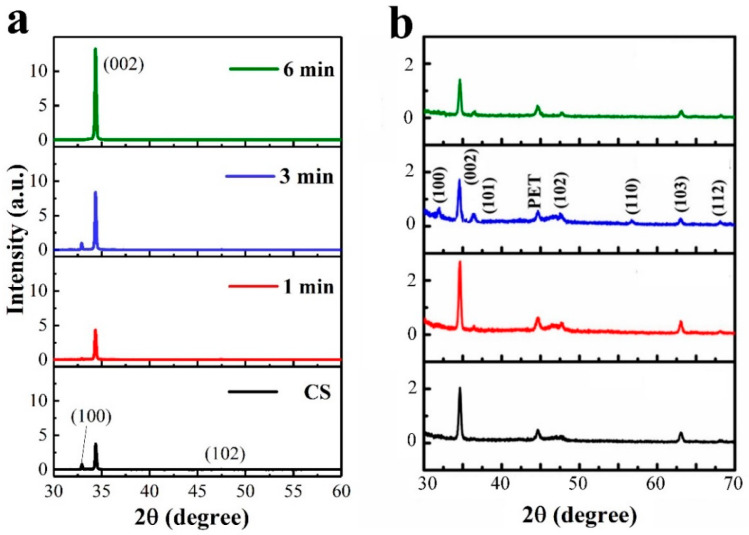
The 2-theta scan patterns recorded from the X-ray diffraction of ZnO NRs with as-grown SLs (CS) and the SLs N_2_O plasma-treated for 1, 3, and 6 min in the cases of growths on (**a**) the GaN and (**b**) PET substrates.

**Figure 6 nanomaterials-11-02011-f006:**
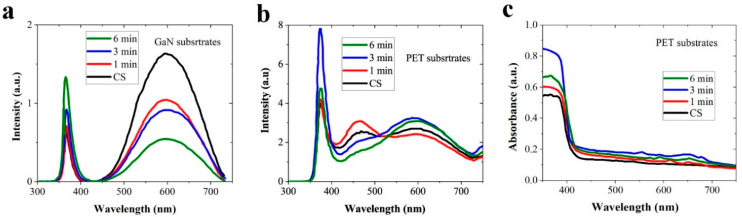
Room temperature PL spectra of ZnO NRs with as-grown SLs (CS) and the SLs N_2_O plasma-treated for 1, 3, and 6 min in the cases of growths on (**a**) the GaN and (**b**) PET substrates. (**c**) UV/visible spectra of the NRs measured at room temperature.

**Figure 7 nanomaterials-11-02011-f007:**
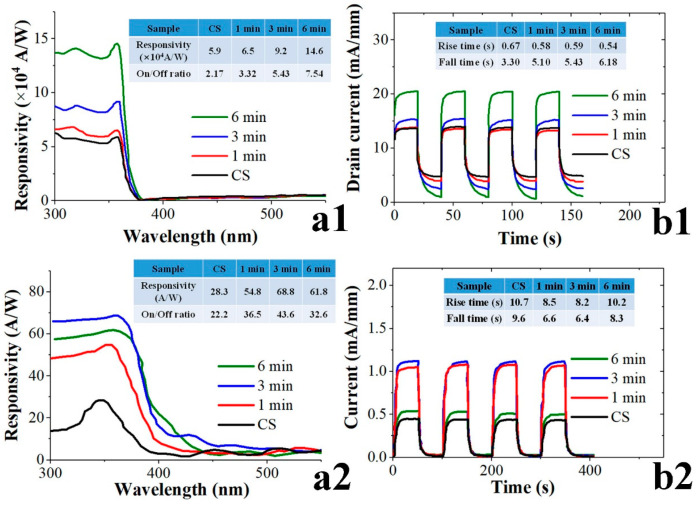
Spectral responsivities and transient characteristics of (**a1**,**b1**) the HEMT-based PDs and (**a2**,**b2**) IDE pattern-based UV PDs fabricated on the PET substrates.

**Figure 8 nanomaterials-11-02011-f008:**
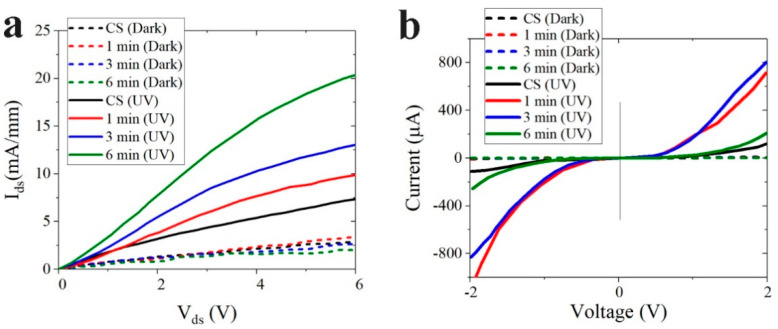
*I-V* characteristics of (**a**) the HEMT-based and (**b**) IDE pattern-based UV PDs.

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
