# Peer review of "Effects of Seed-Layer N2O Plasma Treatment on ZnO Nanorod Based Ultraviolet Photodetectors: Experimental Investigation with Two Different Device Structures"

_nanomaterials, 2021, doi:10.3390/nano11082011_

Round 1
Reviewer 1 Report
A few additions are necessary to improve the scientific quality of the paper.
- First, in Fig. 1e, it is difficult to understand the corresponding between the drawing and the fabricated device in orange color.
- unit (nm) in Fig. 3d is missing.
-Fig. 3 reports summary of mean diameters, average lengths, and aspect ratios (length/diameter) of ZnO NRs grown under different SL treatment condition on the GaN and PET substrates. In the case of GaN substrate, the trends seem to be coherent as a function of SL treatment. How to explain the maximum value of aspect ratio for the PET substrate at 3mn ?
- In fig.5, I suppose that PL spectra were performed at 300K. Can these measurements be coupled to absorption measurement ?
- For the PET substrate sample (Fig. 5b), PL peaks appearing at higher wavelength are attributed to NRs intrinsic defect states. Why such signatures are not present for the GaN substrate (Fig. 5a) ?
- the authors said that I-V characteristics were measured. Why these measurements are missing in the paper ?
- I'm very surprised by the responsivity values, with values higher than 1.5E5 A/W and 60 A/W for the HEMT-based and IDE pattern-based devices, respectively. What are the quantum efficiency of these PDs ? For these measurements, why a bias voltage of 2V is applied ; how the incident flux is controlled ?
- in order to assess the performance achieved, comparison with commercial UV detectors is necessary.
Reviewer 2 Report
Lee et al, have studied the effects of seed-layer N2O plasma treatment on the performance of UV photodetectors based on ZnO nanorods. They found suitable plasma treatment can improve the performance of photodetectors due to the increased ZnO crystallinity, which can be evidenced by XRD and PL spectra. However, some issues still need to be addressed and some typos need to be corrected.
- Figure caption of Fig. 5 is wrong (the same as Fig. 4).
- In addition to PL spectra, time-resolved PL decay curves should provide to confirm the improvement of the crystallinity of ZnO by seed-layer plasma treatment.
- It is more convenient for the readers by drawing a scheme to illustrate the operational processes in both photodetectors.
- Representative I-V curves for both photodetectors should be included in the context.
Round 2
Reviewer 1 Report
Comments and remarks have been considered in the new version
Reviewer 2 Report
The authors have addressed my questions and the manuscript has been improved.